# Large Rab GTPases: Novel Membrane Trafficking Regulators with a Calcium Sensor and Functional Domains

**DOI:** 10.3390/ijms22147691

**Published:** 2021-07-19

**Authors:** Takayuki Tsukuba, Yu Yamaguchi, Tomoko Kadowaki

**Affiliations:** 1Department of Dental Pharmacology, Graduate School of Biomedical Sciences, Nagasaki University, Sakamoto 1-7-1, Nagasaki 852-8588, Japan; yu-y@nagasaki-u.ac.jp; 2Department of Frontier Oral Science, Graduate School of Biomedical Sciences, Nagasaki University, Sakamoto 1-7-1, Nagasaki 852-8588, Japan; tomokok@nagasaki-u.ac.jp

**Keywords:** Rab GTPase, Rab44, Rab45/RASEF, Rab46, EF-hand domain, coiled-coil domain, pro-line-rich domain

## Abstract

Rab GTPases are major coordinators of intracellular membrane trafficking, including vesicle transport, membrane fission, tethering, docking, and fusion events. Rab GTPases are roughly divided into two groups: conventional “small” Rab GTPases and atypical “large” Rab GTPases that have been recently reported. Some members of large Rab GTPases in mammals include Rab44, Rab45/RASEF, and Rab46. The genes of these large Rab GTPases commonly encode an amino-terminal EF-hand domain, coiled-coil domain, and the carboxyl-terminal Rab GTPase domain. A common feature of large Rab GTPases is that they express several isoforms in cells. For instance, Rab44’s two isoforms have similar functions, but exhibit differential localization. The long form of Rab45 (Rab45-L) is abundantly distributed in epithelial cells. The short form of Rab45 (Rab45-S) is predominantly present in the testes. Both Rab46 (CRACR2A-L) and the short isoform lacking the Rab domain (CRACR2A-S) are expressed in T cells, whereas Rab46 is only distributed in endothelial cells. Although evidence regarding the function of large Rab GTPases has been accumulating recently, there are only a limited number of studies. Here, we report the recent findings on the large Rab GTPase family concerning their function in membrane trafficking, cell differentiation, related diseases, and knockout mouse phenotypes.

## 1. Introduction

Intracellular membrane trafficking involves complicated network systems that precisely regulate various membrane-bound organelles in cells [1,2]. Dynamic membrane movements, including vesicle transport, membrane fission, tethering, docking, and fusion events are controlled by Rab GTPases, which act as central regulators [3,4]. Rab GTPases are strictly controlled by two important regulators: the guanine nucleotide exchange factors (GEFs) and GTPase-activating proteins (GAPs) that undergo conformational changes by translocation between the cytosol and membranes [5,6]. GEFs change the inactive GDP-bound Rab protein to the active GTP-bound form on the membranes, whereas GAPs convert active GTP-bound Rab protein to the inactive GDP-bound form [7,8,9]. Posttranslational isoprenylation of one or two cysteine residues at the carboxy terminus is required for the membrane association of Rab GTPases [10,11].

To date, approximately 70 members of Rab GTPases have been discovered in mammalian cells [12,13,14]. Rab GTPases are roughly divided into two groups: conventional “small” Rab GTPases and atypical “large” Rab GTPases. The former group includes Rab 1–43 with low molecular weights of approximately 20–30 kDa, which are well-characterized [15,16,17]. Among these, some are known as “housekeeping Rabs”, since they are conserved in eukaryotes [4,12]. Some examples are Rabs1, 2, and 6 that act in the Golgi; Rab8 is involved in Golgi-to-plasma membrane trafficking; and Rabs 4, 5, 7, 11, 14, and 21 are implicated in the endosomal system [18]. Furthermore, some Rab proteins play a specific role in animal cells: Rabs 3, 26, 27, 33, 37, and 39, are mainly involved in regulating secretion; Rabs 10 and 43 are involved in membrane trafficking between the Golgi and ER; Rabs 30, 33, 34, and 43 play a role in Golgi localization; and Rabs9 and 22 are involved in late endocytic trafficking [12,18].

In contrast, large Rab GTPases comprise Rab44, Rab45 (alias RASEF (RAS and EFD-containing protein)), and Rab46. These large Rab GTPases are atypical Rab GTPases with molecular weights of approximately 70–150 kDa, which have been reported recently. However, studies are limited regarding these GTPases thus far.

In this review, we report the recent findings on the large Rab GTPase family, including their function in membrane trafficking, cell differentiation, related diseases, and knockout mouse phenotypes.

## 2. Common Features of Large Rab GTPases

One of the important features of large Rab GTPases is that these proteins share a common structural domain organization that includes an amino-terminal EF-hand domain (EFD), coiled-coil domain (CCD), and a carboxy-terminal Rab domain (Figure 1). EFDs are Ca^2+^ binding sites with a helix-loop-helix topology [19,20,21]. For example, the EFD of Rab44 was suggested to be required for its translocation from the lysosomes to the plasma membrane and/or cytosol following transient Ca^2+^ influx [22]. Rab45 and Rab46 contain two EFDs (Figure 1). The second EFD of Rab46 reportedly binds to Ca^2+^ [23,24]. The EFDs of Rab45 have not been investigated. In the case of Rab46, a mutant lacking the EFD of Rab46 loses its interaction with dynein–dynactin, although this interaction is independent of Ca^2+^ concentration [25].

Generally, CCD-containing proteins have been proposed to act as tethering factors to target organelles before fusion, or as scaffolds for the assembly of other factors important for fusion [26,27]. The CCD of Rab44 appears to be important for localization and organelle formation [22]. In particular, Rab44 mutants lacking the CCD are efficiently colocalized with the lysosomes; however, they show reduced colocalization with the ER [22]. Moreover, expression of Rab44 mutants lacking the CCD affects the lysosomal size compared to that of the wild type [22]. Furthermore, the CCD of Rab45/RASEF is important for self-interaction and oligomer formation [28,29]. Interaction analyses using the yeast two-hybrid system indicated that the CCD of Rab45/RASEF specifically associates with the full-length or the CCD of Rab45/RASEF, but not with the Rab domain or the EFD [28].

Proline-rich domains (PRDs) include specific proline-containing sequences of three to six residues [30] that are important for protein–protein interactions with low affinities and moderate specificities [31,32]. Although a previous report suggested that all members of large Rab GTPases contain PRDs [33], we propose that Rab44 and Rab46 contain the PRD, but Rab45 lacks it. The amino acid sequence between the CCD and Rab domain of large Rab GTPases is shown in Figure 2. Some consensus sequences containing proline have been identified in these regions of Rab44 and Rab46, but not in Rab45 (Figure 2). Since these domains that recognize target core motifs contain phosphorylated residues [30,34], this domain in large Rab GTPases is thought to function as a switch between binding partners to control association and dissociation in the phosphorylated state. The PRD of Rab46 interacts with the GEF protein Vav1, which is converted into a tyrosine-phosphorylated protein upon T cell receptor (TCR) stimulation [33,35]. Vav1 binding mediated by its PRD is due to the SH3–SH2–SH3 domain in the carboxy-terminus of Vav1 [35]. Therefore, the Rab46-mediated interaction with Vav1 is thought to be required for the recruitment of Rab46-containing vesicles to the immunological synapse in T cells [33,35].

Another common feature of large Rab GTPases is the existence of isoforms generated by splicing variants. Mouse Rab44 is expressed as two isoforms: Rab44 long form (Rab44-L) that encodes 924 amino acids, and Rab44 short form (Rab44-S) that comprises 725 amino acids [36,37] (Figure 1). Mouse Rab44-L consists of EFD, CCD, PRD, and the Rab domain, although 47 amino acids in the CCD and 38 amino acids in the region between CCD and Rab domain are absent compared to the human Rab44 (Figure 1). Furthermore, mouse Rab44-S lacks the EFD and N-terminal half of the CCD (Figure 1). In humans, only the long form of Rab44 (Rab44-L) was reported, and Rab44-S has not yet been discovered [22] (Figure 1). Similar to Rab44, mouse Rab45/RASEF is expressed as two isoforms: the long form (Rab45-L) that encodes 627 amino acids [28], and the short form (Rab45-S) that encodes 530 amino acids and is predominantly present in the testes [29] (Figure 1). The mouse Rab45-L consists of EFD, CCD, and the Rab domain, whereas Rab45-S contains CCD and the Rab domain but lacks the N-terminal EFD. CRACR2A/Rab46 comprises two isoforms: the short form CRACR2A-S (alias: CRACR2A-c) consisting of 395 amino acids, containing only the EF-hand and CCDs, and without the Rab domain [23,33] (Figure 1); and the long form (CRACR2A-L, alias: CRACR2A-a), which was identified as a protein composed of 732 amino acids in human umbilical vein endothelial cells [38] and subsequently found in T cells [35]. The Rab46 long form has a Rab domain and an isoprenylation site at the carboxy terminus.

A previous study identified large Rab GTPase orthologs for the nematode *Caenorhabditis elegans* Rab proteins 4R79.2 and rsef-1, indicating the evolutionarily conserved nature of Rab proteins (Figure 1). The predicted Rab44 ortholog 4R79.2 (alias: CeRabY2) expresses two isoforms, 4R79.2a, with 395 amino acids, and 4R79.2b, with 311 amino acids [39,40] (Figure 1). The former contains an EFD and a Rab domain, whereas the latter lacks EFD but possesses the Rab domain. The *rsef-1* (alias: C33D12.6/TAG-312/CeRabY1) gene encodes an EFD, CCD, and Rab domain (Figure 1). However, the existence of isoforms of Rab45/RASEF ortholog *rsef-1* in *C. elegans* has not yet been reported [41] (Figure 1). 

## 3. Rab44

Rab44 was originally identified as an upregulated gene during osteoclast differentiation [42]. Subsequent tissue distribution analysis of mouse Rab44 indicated that Rab44 is highly expressed in the bone marrow and marginally expressed in the epididymis, lungs, skin, spleen, thymus, ovary, uterus, and liver [37]. In the bone marrow, Rab44 is extensively expressed in hematopoietic CD117^+^ Sca-1^−^ cells, which are granulocyte- or mast cell-lineage cells. Therefore, Rab44 is highly expressed in immune-related cells, including mast cells, bone marrow cells, and osteoclasts [37]. 

### 3.1. Intracellular Localization

Endogenous mouse Rab44 localizes mainly to the lysosomes and the ER and partially to early endosomes in bone marrow mast cells [36]. In particular, Rab44 isoforms exhibit differential localization. The expression of the human Rab44-L and the two mouse isoforms was examined in rat basophilic leukemia (RBL)-2H3 cells. The results indicated that both human and mouse Rab44-L localize mainly in the lysosomes, whereas the mouse Rab44-S localizes mainly in the ER [36]. In mouse macrophages, overexpressed Rab44-S was localized in the lysosomes and the Golgi complex [42], suggesting that the localization of Rab44 is cell-specific. When human Rab44-L was expressed in HeLa cells, Rab44 formed ring-like structures, and partially surrounded the lysosomes in the cells [22]. 

Rab44 localization in the lysosomes is partially altered to the plasma membrane and cytosol upon transient Ca^2+^ influx. The partial translocation of Rab44 is required for the EFD, since mutants lacking EFD fail to translocate [22]. Moreover, the CCD appears to be important for localization and organelle formation.

### 3.2. Cell Function

Rab44 negatively regulates osteoclast differentiation. Rab44 knockdown promotes osteoclast differentiation of macrophages, and conversely, Rab44 overexpression prevents osteoclast differentiation [42]. Mechanistically, Rab44 specifically regulates lysosomal Ca^2+^ influx and then affects the nuclear factor of activated T cells c1 (NFATc1) signaling in RANKL-stimulated macrophages [42].

Rab44 regulates degranulation of mast cells. Mast cells derived from Rab44-deficient mice show reduced FcεRI-mediated β-hexosaminidase secretion compared to mast cells from wild-type mice [36,43]. Specifically, Rab44-deficient mice show reduced anaphylactic responses in vivo [43]. Moreover, although Rab44 isoforms exhibit differential localization in mast cells, they promote lysosomal exocytosis in a similar manner [36]. Expression of the human Rab44-L and both mouse isoforms causes an increase in β-hexosaminidase secretion in rat RBL-2H3 cells [36].

### 3.3. Binding Molecules and/or Effectors

Both human and mouse Rab44 proteins interact with vesicle-associated membrane protein 8 (VAMP8), a v-SNARE protein [36].

### 3.4. Knockout Phenotypes or Diseases

Rab44 deficiency is likely to be associated with allergies and autoimmune diseases. Rab44 knockout mice grow normally, and exhibit reduced systemic anaphylaxis as well as lower histamine levels in the serum than wild-type mice [36,43]. 

Whole-genome sequencing of samples from patients with T-cell-mediated autoimmune lymphoproliferative disease revealed that Rab44 is one of the missense genes [44]. A transcriptome-wide association study indicated that Rab44 was upregulated in white blood cells from patients with atopy [45]. 

The distribution, localization, functions, and binding proteins of Rab44 and the phenotypes or diseases associated with Rab44 deficiency are summarized in Table 1.

## 4. Rab45/RASEF

Rab45/RASEF was initially discovered as a downregulated gene in human cutaneous malignant melanoma by genome-wide screening analysis [46]. Subsequent gene expression analyses have shown that human Rab45/RASEF is detected in many cancer cells, such as myeloid leukemia [47], uveal malignant melanoma [48], colorectal cancer [49], lung cancer [50], and breast cancer [51,52,53]. Moreover, Rab45/RASEF has been found in normal cells, including airway epithelial cells, columnar epithelial cells, bronchial epithelial cells [50], and non-cancerous breast epithelial cell lines [51]. Therefore, Rab45/RASEF is likely to be expressed predominantly in various epithelial cells, either normal or malignant. Immunohistochemical analysis using normal human tissues indicated that Rab45/RASEF is detectable in the liver, heart, kidney, lungs, prostate, and testes [50].

Rab45/RASEF is likely to be induced by certain stimuli. Rab45/RASEF expression is regulated by promoter methylation of the *Rab45/RASEF* gene. Cigarette smoking induces DNA methylation in the promoter region of the *Rab45/RASEF* gene in arterial smooth muscle cells [54]. A mutation in the *Rab45/RASEF* promoter region causes methylation and low RASEF expression in uveal melanoma, suggesting that Rab45/RASEF acts as a tumor suppressor gene [48]. Interestingly, the short form of Rab45/RASEF (Rab45/RASEF-S) is an androgen-induced protein in the testes [29]. Rab45/RASEF-S is also distributed in germ cells, particularly in elongated spermatids [29]. 

### 4.1. Intracellular Localization

Rab45/RASEF was reported to have differential localization in different cell types. Human Rab45/RASEF-L expressed in HeLa cells predominantly localizes in the Golgi apparatus and/or recycling endosomes [28]. For localization, both the CCD and Rab domain are required, but not the EFD. In particular, oligomer formation by the CCD of Rab45/RASEF appears to be important for localization, since a CCD deletion mutant showed dispersed localization patterns near the perinuclear region [28]. In human lung cancer cells, endogenous Rab45/RASEF localizes mainly in the cytoplasm [50]; however, the localization mechanism is still unknown. Testis-specific mouse Rab45/RASEF-S was reported to contain a nuclear targeting sequence in the middle of the CCD, despite the presence of a prenylation site at the carboxy-terminus [29]. When mouse Rab45/RASEF-S was exogenously expressed in HeLa cells, it was localized in the nuclei [29]. However, a deletion mutant lacking the nuclear targeting sequence of Rab45/RASEF-S showed cytoplasmic localization [29]. Thus, a more detailed analysis is required for the nuclear or cytosolic targeting of Rab45/RASEF.

### 4.2. Cell Function

The function of Rab45/RASEF in cancer cells, either as a tumor suppressor or oncogene, is controversial. Rab45/RASEF was thought to act as a tumor suppressor, since it was initially discovered as a downregulated gene in human cutaneous malignant melanoma [46]. Overexpression of RAB45/RASEF caused apoptosis by activation of caspase-3 and -9 and increased phosphorylation of p38 in chronic myelogenous leukemia cells [55]. Moreover, a clinical study showed that cancer patients with low Rab45/RASEF expression levels had significantly lower survival rates than patients with high Rab45/RASEF expression levels, suggesting that Rab45/RASEF may be a tumor suppressor gene [49].

In contrast, Rab45/RASEF was reported to act as an oncogene in some cancer cells. RASEF overexpression promoted cell growth, whereas Rab45/RASEF knockdown reduced the growth of cancer cells [50]. Mechanistically, Rab45/RASEF-L interacts with extracellular signal-regulated kinase (ERK) 1/2 and enhances ERK1/2 signaling [50]. Importantly, inhibiting the interaction between Rab45/RASEF-L and ERK1/2 using a cell-permeable peptide that corresponded to the ERK1/2-interacting site of RASEF suppressed the growth of lung cancer cells.

### 4.3. Binding Molecules and/or Effectors

Co-transfection with FLAG-Rab45 and Myc-Rab45, and subsequent immunoprecipitation with anti-Myc antibody followed by Western blotting with anti-FLAG antibody revealed that both Rab45/RASEF-L or -S (a truncation mutant) interact with each other, indicating oligomer formation [28,29]. Rab45/RASEF interacted with ERK1/2 [50]. An immunoprecipitation assay using anti-RASEF antibody revealed the interaction between endogenous RASEF and endogenous ERK1/2 proteins in lung cancer cells [50]. Recently, using an in vitro pull-down assay, recombinant Rab45/RASEF was reported to be associated with the motor protein dynein–dynactin complex [25], suggesting microtubule minus end-directed transport.

### 4.4. Knockout Phenotypes or Diseases

The *Rab45/RASEF* gene is encoded on the long arm of chromosome 9, where the region is frequently deleted in candidate tumor suppressor genes in patients with acute myeloid leukemia [47]. To date, *Rab45/RASEF* knockout mice or transgenic mice have not been reported.

The nematode *Caenorhabditis elegans*, a model animal, expresses the Rab45/Rasef ortholog *rsef-1* gene in uterine seam cells (Figure 1). The rsef-1-GFP protein is expressed in both the spermathecal and spermathecal-uterine junction at an earlier stage, and in the uterine toroids at young adult [41]. Knockdown of the *rsef-1* gene with siRNA in *C. elegans* causes defects in nuclear translocation and cell proliferation in endometrial epithelial cells [41].

The distribution, localization, cell function, and binding proteins of Rab45/RASEF are summarized in Table 1.

## 5. Rab46 (CRACR2A-L or CRACR2A-a)

The gene encoding Rab46 was originally designated *CRACR2A* (Ca^2+^ release-activated Ca^2+^ channel regulator 2A) or *EFCAB4B* (EF-hand calcium binding domain 4B). The CRACR2A protein, which contains 395 amino acids, was first identified as a regulator of store-operated calcium entry in T cells [23]. The isoform was described as CRACR2A-S (alias CRACR2A-c), since this protein is a short form lacking a Rab domain. A subsequent study revealed a longer variant CRACR2A-L (also known as CRACR2A-a), which contains 731 amino acids in endothelial cells [38]. The long isoform was renamed Rab46 since it contains a Rab domain [56]. Therefore, the short isoform CRACR2A-S is not discussed in this review.

Rab46 is reportedly expressed in endothelial cells [38] as well as immune cells including T cells [35]. The mouse database of the tissue distribution of the *CRACR2A* gene indicates that it is distributed in lymphoid tissues, such as the thymus, spleen, and other tissues, including the lungs and large and small intestines (https://www.ncbi.nlm.nih.gov/gene/381812/?report=expression, accessed on 13 July 2021). However, whether the data are derived from CRACR2A-S or Rab46 is unknown.

### 5.1. Intracellular Localization

Localization of Rab46 is likely to alter in resting and stimulated cells. Rab46 predominantly localizes in the Golgi in resting T cells. GTP binding is necessary for the correct localization of Rab46. After TCR stimulation, Rab46 is immediately translocated to specialized vesicles formed by Vav1, which leads to the activation of the downstream JNK pathway [35]. The PRD is important for the recruitment of Rab46-containing vesicles into the immunological synapse.

In endothelial cells, Rab46 is initially localized to Weibel-Palade bodies (WPBs), which are small storage granules containing von Willebrand factor and P-selectin [56]. Upon histamine stimulation, but not thrombin stimulation, Rab46 is directly translocated from WPBs to the perinuclear region, possibly the Golgi [56].

### 5.2. Cell Function

In CD4^+^ T cells, Rab46 controls the Orai1-dependent Ca^2+^ influx during TCR signaling, which induces Ca^2+^/NFAT-and JNK/AP1-dependent pathways [57]. Although both CRACR2A-S and Rab46 share a conserved function in the Ca^2+^-NFAT signaling pathway, only Rab46 is involved in the activation of the JNK pathway, suggesting the importance of the PRD and Rab domains in this functional difference [35]. Moreover, Rab46-mediated signaling selectively affects Th1 differentiation, which is directly involved in autoimmune diseases [57]. Rab46 functions as a dynein adaptor and engages in Ca^2+^-regulated endocytic trafficking in T cells as observed in an in vitro assay [25]. However, Rab46 in endothelial cells fails to function in the store-operated Ca^2+^ mechanism [38].

In endothelial cells, Rab46 is likely to be involved in tube formation. Rab46 knockdown suppresses endothelial tube formation, especially by reducing lumen length and the number of branch points in endothelial cells [38]. Moreover, in endothelial cells, Rab46 diverts WPBs that contain cargo superfluous to histamine stimulation away from the plasma membrane to the microtubule organizing center (MTOC), thus preventing secretion [56]. This trafficking is independent of Ca^2+^. However, Ca^2+^ binding to Rab46 is necessary at the MTOC for re-dispersal of WPBs after stimulation [56].

### 5.3. Binding Molecules and/or Effectors

The PRD of Rab46 associates with the SH3–SH2–SH3 domains of Vav1, wherein Vav1 is tyrosine-phosphorylated after TCR stimulation [35]. Recently, it has been found that Rab46 directly binds to the motor protein dynein–dynactin complex [24,25,58], and the Na^2+^/K^+^ ATPase subunit alpha 1 in endothelial cells [24].

### 5.4. Knockout Phenotypes or Diseases

When T-cell-specific *Rab46* gene (*CRACR2A*) knockout mice were generated, Rab46 deficiency in primary T cells impaired the activation of the Ca^2+^-NFAT and JNK signaling pathways [35]. Moreover, Rab46 deficiency impairs Th1 differentiation in vitro [57]. Consistent with these findings, Rab46 depletion in T cells decreased Th1 responses in acute lymphocytic choriomeningitis virus infection and reduced resistance to experimental autoimmune encephalomyelitis in vivo [57].

A genome-wide association study indicated that four single-nucleotide polymorphisms in CRACR2A/Rab46 (EFCAB4B) caused periodontal diseases [59], and nonalcoholic fatty liver (NAFL) and/or nonalcoholic liver disease (NAFLD) [60,61,62,63]. Genetic analysis of the human genome indicated that Rab46 is an inducible protein, since exercise causes hypomethylation of the promoter region of the *CRACR2A* gene, resulting in lower risk of metastatic prostate cancer [64] and lung cancer [65].

The distribution, localization, functions, and binding proteins of Rab46 and the phenotypes or diseases associated with CRACR2A/Rab46 deficiency are summarized in Table 1.

## 6. Concluding Remarks and Perspectives

In this article, we summarized the recent findings on large Rab GTPases, including Rab44, Rab45/RASEF, and Rab46. As large Rab GTPases contain an EFD, it is speculated that regulation of the Ca^2+^ signaling pathway is important for large Rab GTPases. Although large Rab GTPases contain many phosphorylation sites, the responsible kinase(s) have not yet been identified. However, small Rab GTPases are phosphorylated by many kinases, such as CDK1, LRRKs, PKCs, TAK1, TBK1, and Src [66]. Therefore, the identification of large Rab GTPase-mediated mechanisms involved in regulating signaling pathways and membrane trafficking is critical. Thus, it is crucial to elucidate the link between membrane trafficking and signaling pathways mediated by large Rab GTPases.

The detailed mechanisms of the functional similarities between large and small Rab GTPases are unclear. For example, both Rab44 and Rab27B colocalize in secretory granules and regulate mast cell degranulation. Mast cells from *Rab44*-deficient mice exhibited decreased release of secretory granules. However, small amounts of secretion from *Rab44*-deficient mast cells have been detected, and therefore, it is assumed that Rab27B is possibly responsible for the alternative secretion of secretory granules. This is because Rab27B-deficient mast cells also display phenotypes similar to those of Rab44 [67,68]. Thus, Rab44 may regulate the release of secretory granules from mast cells independent of Rab27B. However, mechanisms other than this are still unknown.

Taken together, elucidating the signaling mechanisms mediated by these large Rab GTPases, as well as investigating the functional similarities between large and small Rab GTPases may potentially unravel new aspects of membrane trafficking.

## Figures and Tables

**Figure 1 ijms-22-07691-f001:**
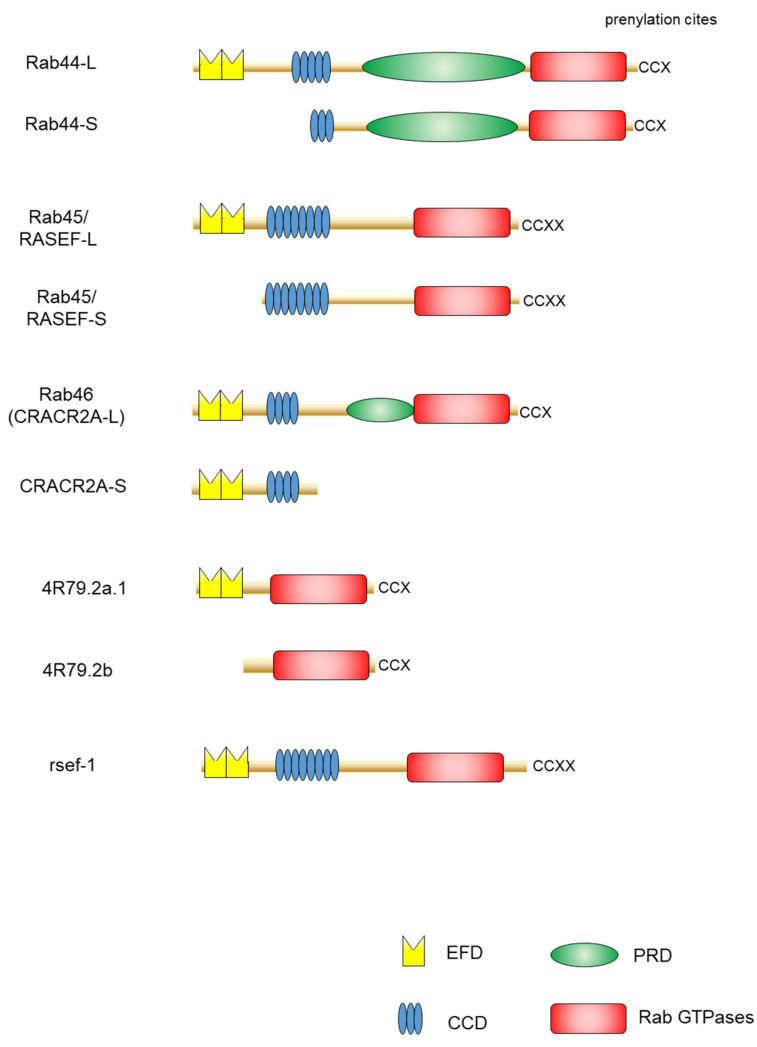
Schematic representation of domain structures of large Rab proteins. The EF-hand domain (EFD), coiled-coil domain (CCD), proline-rich (PRD), and the Rab family domain are shown in yellow, blue, green, and red, respectively.

**Figure 2 ijms-22-07691-f002:**
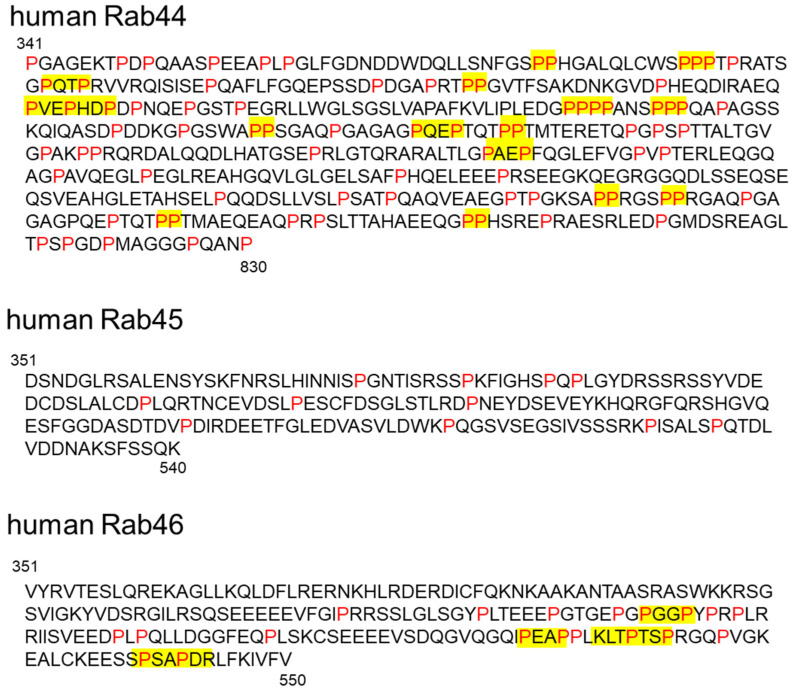
Sequence alignment of regions between the coiled-coil domain and the Rab domain in large Rab GTPases. The red letters indicate proline residues. Yellow indicates proline-rich sequences, including PxxPxxP, PPxPPxPP, PxxP, (R/K)xxPxxP, PxxPx(R/K), and xPPx. Accession numbers for the proteins are the following: human Rab44 (NP_001244286.1), human Rab45/RASEF (NP_689786.2), and human Rab46 (NP_001138430.1).

**Table 1 ijms-22-07691-t001:** Summary of distribution, localization, functions, binding proteins, and the phenotypes or diseases associated with large Rab GTPases.

Protein Name (Localization)	Distribution	Functions	Binding Molecules	Phenotypes
Rab44-L				
	Mast cells	Degranulation	VAMP8	Reduced anaphylaxis
	Bone marrow			Autoimmune diseases
	Immune cells			Atopic diseases
	Osteoclasts (ER/L)	Osteoclast-differentiation		
Rab44-S				
	Osteoclasts	Osteoclast-differentiation		
	Mast cells	Degranulation	VAMP8	
	Bone marrow			
	Immune cells (ER/G/L/EE)			
Rab45/RASEF-L				
	Liver, heart, kidney, lung, prostate, testis		Rab45/RASEF (oligomer)	
	Epithelial cells		Dynein–dynactin	
	Cancer cells (G/RE/C)	Tumor suppressor or oncogene	ERK1/2	
Rab45/RASEF-S	Germ cells of testis (N)		Rab45/RASEF (oligomer)	
Rab46				
	T cells (G/Vav1^+^-vesicles)	Transport of Vav1^+^-vesicles	Vav1	Impaired Th1 differentiation
	Endothelial cells (Tube formation)	Transport of WPBs	Dynein–dynactin	Periodontal diseases Cancer metastasis
			Na^2+^/K^+^ ATPase subunit α1	NAFL, NAFLD

Abbreviations: ER: endoplasmic reticulum, G: Golgi complex, L: lysosomes, EE: early endosomes, RE: recycling endosomes, N: nuclei, PM: plasma membrane, C: cytosol, WPBs: Weibel–Palade bodies.

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
