# Peer review of "Large Rab GTPases: Novel Membrane Trafficking Regulators with a Calcium Sensor and Functional Domains"

_ijms, 2021, doi:10.3390/ijms22147691_

Round 1

Reviewer 1 Report

In this review, the authors surveyed and compiled literature on studies covering unconventional members of Rab GTPases, including Rab44, Rab45/RASEF and Rab46/CRACR2A-L. The review describes what is known about their expression, localization, function, interacting partners and effects following perturbation of their function and in human diseases, where applicable. Overall, the manuscript is well organized, clearly written and informative. Some concerns to be address: 

1.     Lines 68-69 “although this interaction is independent of either lower or higher Ca2+ concentration”. The words “either lower or higher” should be removed.

2.     Figure 2: References should be made to the numbers referring to amino acid positions in the legend.

3.     Table 1: Formatting issues in the rows for Rab45/RASEF-S (Rab45/RASEF) and Rab46 (Tube formation). Table 1 should have a table title for clarity.

Author Response

Comments and Suggestions for Authors

In this review, the authors surveyed and compiled literature on studies covering unconventional members of Rab GTPases, including Rab44, Rab45/RASEF and Rab46/CRACR2A-L. The review describes what is known about their expression, localization, function, interacting partners and effects following perturbation of their function and in human diseases, where applicable. Overall, the manuscript is well organized, clearly written and informative. Some concerns to be address:

  1. Lines 68-69 “although this interaction is independent of either lower or higher Ca2+ concentration”. The words “either lower or higher” should be removed.

Answer: First of all, we appreciate for positive comments for our paper. According to the reviewer’s suggestion, we have deleted the words “either lower or higher”.

  1. Figure 2: References should be made to the numbers referring to amino acid positions in the legend.

Answer: Thank you for your important comments. The reviewer’s suggestion is correct. We added “Accession numbers for protein are following: human Rab44 (NP_001244286.1), human Rab45/RASEF (NP_689786.2), and human Rab46 (NP_001138430.1).” in the legend of Figure 2.

  1. Table 1: Formatting issues in the rows for Rab45/RASEF-S (Rab45/RASEF) and Rab46 (Tube formation). Table 1 should have a table title for clarity.

Answer: We appreciate for your important suggestion. We have adjusted the format of Table 1 and added the title as follows “Summary of distribution, localization, functions, binding proteins and the phenotypes or diseases associated with large RabGTPases”.

Reviewer 2 Report

In this article, the authors review the atypical "large" Rab GTPases (Rab44, Rab45/RASEF, Rab46) in terms of their role in cellular responses, such as intracellular trafficking, and their relevance to diseases. This paper cites appropriate references to introduce the recent findings on the "large" Rab GTPases and I believe that this article is suitable to be published as a review article in International Journal of Molecular Sciences.

A Minor point:

Replace manu-script with manuscript (Author Contributions).

Author Response

Comments and Suggestions for Authors

In this article, the authors review the atypical "large" Rab GTPases (Rab44, Rab45/RASEF, Rab46) in terms of their role in cellular responses, such as intracellular trafficking, and their relevance to diseases. This paper cites appropriate references to introduce the recent findings on the "large" Rab GTPases and I believe that this article is suitable to be published as a review article in International Journal of Molecular Sciences.

A Minor point:

Replace manu-script with manuscript (Author Contributions).

Answer: First of all, we appreciate for positive comments for our paper. We have replaced the “manu-script” to manuscript.